# EEI-IoT: Edge-Enabled Intelligent IoT Framework for Early Detection of COVID-19 Threats

**DOI:** 10.3390/s23062995

**Published:** 2023-03-10

**Authors:** B. D. Deebak, Fadi Al-Turjman

**Affiliations:** 1Department of Computer Engineering, Gachon University, Gyeonggido, Seongnam 13120, Republic of Korea; 2Artificial Intelligence Engineering Deptartment, AI and Robotics Institute, Near East University, Mersin 10, Turkey; 3Research Center for AI and IoT, Faculty of Engineering, University of Kyrenia, Mersin 10, Turkey

**Keywords:** COVID-19, internet of things, Edge, IoT technologies, power consumption

## Abstract

Coronavirus disease 2019 (COVID-19) has caused severe acute respiratory syndrome coronavirus 2 (SARS-CoV-2) across the globe, impacting effective diagnosis and treatment for any chronic illnesses and long-term health implications. In this worldwide crisis, the pandemic shows its daily extension (i.e., active cases) and genome variants (i.e., Alpha) within the virus class and diversifies the association with treatment outcomes and drug resistance. As a consequence, healthcare-related data including instances of sore throat, fever, fatigue, cough, and shortness of breath are given due consideration to assess the conditional state of patients. To gain unique insights, wearable sensors can be implanted in a patient’s body that periodically generates an analysis report of the vital organs to a medical center. However, it is still challenging to analyze risks and predict their related countermeasures. Therefore, this paper presents an intelligent Edge-IoT framework (IE-IoT) to detect potential threats (i.e., behavioral and environmental) in the early stage of the disease. The prime objective of this framework is to apply a new pre-trained deep learning model enabled by self-supervised transfer learning to build an ensemble-based hybrid learning model and to offer an effective analysis of prediction accuracy. To construct proper clinical symptoms, treatment, and diagnosis, an effective analysis such as STL observes the impact of the learning models such as ANN, CNN, and RNN. The experimental analysis proves that the ANN model considers the most effective features and attains a better accuracy (~98.3%) than other learning models. Also, the proposed IE-IoT can utilize the communication technologies of IoT such as BLE, Zigbee, and 6LoWPAN to examine the factor of power consumption. Above all, the real-time analysis reveals that the proposed IE-IoT with 6LoWPAN consumes less power and response time than the other state-of-the-art approaches to infer the suspected victims at an early stage of development of the disease.

## 1. Introduction

At present, the globe is facing a health crisis in monitoring and controlling the dispersal of COVID-19. The situation has worsened with global impacts including social facets, economic, education, healthcare systems, etc. Since the rate of transmission is increasing steadily, stakeholders including government and non-government officials have collaborated on the discovery of innovative elements such as screening, contact tracing, testing, isolating, and medical diagnostic [1]. Most of the real-time scenarios involve Industry 4.0 to automate the collection of real-time data to monitor people’s activities. It has an intelligent design in the smart factories to collect the automation data that improves productivity and meets customers’ requirements. However, it is still challenging to address issues such as detection and prediction to provide a better decision-making process. Health impairment is highly contractable in the respiratory droplets of the infected person. Minor symptoms include fever, dry cough, headache, sore throat, loss of taste, and tiredness whereas major symptoms may include organ failure, chest pain, blood clots, kidney injury, and pneumonia. Since there is no proven procedure or vaccine to treat COVID-19, assistive devices and technologies such as care and service support are highly preferred to flatten the data curve or slow the spread of the disease [2].

Most countries including the United States, the United Kingdom, Germany, etc. have experienced several control measures such as contact tracing, testing, quarantine, social distancing, face mask, and lockdowns. However, the number of COVID-19 cases continues to climb higher. As a result, frontline medical experts are increasingly overwhelmed with regards to managing diagnostics, treatment, quarantine centers, and containment zones. To automate the process of virus detection, health intervention workers have developed a self-diagnostic tool [3]. It actively uses saliva swabs to inspect the transmission rate of active cases, or may use a finger prick test to observe the antibodies of the undetected cases. However, with the available test kits it is still challenging to design any predictive modeling to provide a possible test case report. As a result, researchers have explored conventional artificial intelligence (AI) models to check the possible feature of a time crisis [4]. It includes possible criteria to detect and predict the possible active cases of COVID-19. Also, COVID-19 transforms its mutation into various forms through genetic characteristics which predict transmissibility, immunity, and diagnosis. As of now, ~80% of the active cases have been reported as mild-to-moderate without rehabilitation [5].

The disease has common symptoms such as fever, cough, and tiredness which appear to be mild in infected people. On the other hand, the severity of symptoms such as trouble breathing, constant pain in muscle, etc. leads to a few complications such as pneumonia. To determine the infectious disease at an early stage, common medical diagnostic involves various learning technologies (namely AI, Virtual Agents, robotic process automation, and deep learning) [6]. A clinician then applies an appropriate diagnostic tool (i.e., chest X-ray and computed tomography) to diagnose the disease accurately. Moreover, mass surveillance uses drone technology to survey infected zones via an application tracker to initiate digital tracking and physical monitoring. Furthermore, the application trackers deploy an AI platform to accelerate diagnostics and treatment, which can promote health services to solve the challenges of the healthcare industry such as data security and information overhead [7]. Most applications use data retrieval to regulate health crises, improve test accuracy, and speed up clinical specificity. An AI platform utilizes statistical and machine learning models to carry out various computing tasks without explicit communication. Each computing task systematically applies machine-learning techniques to determine the appropriate parameters which can optimize the model to achieve better prediction accuracy [8].

### 1.1. Motivation

COVID-19 has been a contagious disease (i.e., human-to-human via droplet and direct contact ranging from mild to severe illness). The human-to-human transmission reveals the study of severe acute respiratory syndrome which acts as a causative agent of the 2019 coronavirus disease to confirm its practical reference in terms of clinical treatment and diagnosis. To control the transmission rate, health regulation advises people with some proactive guidelines such as wearing face masks, six feet distance, avoiding close contact, and getting vaccinated. These guidelines have a proactive role rather than preventive role for the early prediction of COVID-19 transmission [9]. Since the infection period varies, preventive measures are still being fertile on their research ground not only to regulate the stringent policies but also to predict the COVID-19 infection genuinely. As a result, the healthcare and public health sectors rely on the authentic prediction of future communicable diseases to handle the social and economic disruption caused by COVID-19 [10]. The prediction strategy can even support the government and healthcare administrator to prepare a standard operative procedure and to perform an early identification and prevention of communicable transmission. 

Most healthcare industry extensively handles AI-empowered 5G networks and predictive data analytics to control the transmission of disease [11]. The precise classifications typically involve protective strategies to observe the susceptibilities of COVID-19 that can even classify the risk levels of the patient. Therefore, this paper prefers deep reinforcement learning to manage the computation and communication resources and utilizes the device intelligence at the edge to process the complex computation of a large dataset [12]. It introduces a location-aware system to train the framework that may in turn mitigate the storage complexities to upload the data via a dedicated wireless channel. Moreover, it has a provision of location-awareness to adapt the network environment to any heterogeneous condition that uses a specific application interface to preserve data privacy [13].

### 1.2. Contributions

The propagative features including time and space are highly influenced during the communicable disease of COVID-19 to develop an edge intelligence system. In this system, three significant parameters such as confirmed, death, and recovery cases are included as a regression problem ~10 Days to forecast a few state-of-the-art supervised deep learning models namely ANN, CNN, and RNN [14]. These state-of-the-art models utilize a statistical dataset of COVID-19 which is preprocessed into two distinctive subsets including the training set $\left(80\%\right)$ and testing set $\left(20\%\right)$ [15]. Finally, the performance evaluation considers four significant metrics such as recall or sensitivity, precision, F1-Score, and accuracy to correlate the diagnostic accuracy with the rules of clinical prediction. The major contributions are as follows:Build an intelligent Edge-IoT model to observe the treatment cases such as supportive care, isolation, and experimental measures.Apply the deep learning model with a few technical aspects including SMOTE, RFS, and STL to construct an ensemble-based hybrid learning model which extracts the optimistic features to shorten the prediction time and error rateDesign a practical testbed using a three-tier protocol stack that classifies the selection criteria of the proposed IE-IoT and other existing frameworks [16,17] to examine the significant parameter such as power consumption and average response time over the short-range communication (i.e., BLE, Zigbee, and 6LoWPAN).

### 1.3. Paper Organization

The remaining sections of this paper are as follows: Section 2 discusses the relevant state-of-the-art approaches in the core aspects of IoT and learning mechanisms. Section 3 shows a high-level IE-IoT model to formulate technical aspects of the cognitive-aware system which streamlines with scenarios of convergence networks to provide a better network performance in terms of quality of services. Section 4 presents an vision of the proposed intelligent IoT framework to monitor and analyze the potential threats of coronaviruses in real-time. In addition, this section conducts a rigorous analysis of the COVID-19 dataset to analyze the prediction rate of virus infection using deep learning models such as ANN, CNN, and RNN. Section 5 demonstrates the learning outcomes of the deep learning models using evaluation metrics. Also, this section shows a practical testbed that uses the proposed IE-IoT and other existing frameworks to analyze the system parameters such as power consumption and average response time. Section 6 concludes our research.

## 2. Related Works

This section discusses the relevant studies in the field of IoT, machine learning, and systematic analysis to offer remote monitoring and automatic diagnostic systems. These systems can use statistical and mathematical approaches to detect and prevent communicable diseases such as COVID-19 [18]. The innovative approach can even improve the performance of the susceptible infection recovery (SIR) model to analyze the behavior of the epidemiologic factors which access premature detection of viruses at any vulnerable zone during the period of the epidemic [19]. In this model, an approach named the time series differential equation is employed to inspect the growth rate of the pandemic. However, the prediction rate of the affected people is still challenging due to the uneven distribution of clinical observation, irrelevant features, training, and execution time of the learning algorithms [20]. These challenging issues demand appropriate evaluation models to train and classify the samples into any useful data or information. Further, the evaluation models utilize cross-validation and resampling strategies to detect communicable diseases such as COVID-19 at an early stage to enhance the patient care systems [21]. To observe its effective measurement, the patient care systems isolate the suspected cases of COVID-19 including asymptomatic carriers.

The clinical techniques, namely imaging and radiology, determine a typical stage of COVID-19 which summarizes the severity score of the coronary arteries scaling $\sim \left[0\;\mathrm{to}\;10\right]$ [22]. A metaheuristic framework generalizes mathematical equations such as Jacobi polynomials and Boltzmann distribution to describe the spread of Meta COVID-19 dynamically without any prior statistical data [23]. This former method adopts the daily records as input to discover the possible output in the polynomial model. Moreover, this model uses only one parameter to examine the procedure iteratively. Lastly, the theoretical analysis applies the model of Meta COVID-19 to analyze the clinical features caused by coronavirus disease. To compute the feature weights, the shapely additive explanations (SHAP) examine the parameters of health-risk assessment. The deterministic features include platelets and eosinophils to classify the emergency caused by the new symptoms of coronavirus. The new findings suggest a new classification method using machine learning to infer the severity of COVID-19 and to categorize the risk level of the patient via a fact-analysis program [24]. Jiang et al. [25] applied six different learning models namely k-NN, logistic, regression, decision tree, random forest, and SVM to examine the clinical features of the coronavirus. The analytical results prove that the SVM achieves better accuracy (i.e., ~80%) than the other classifiers.

Alakus and Turkoglu [26] utilized various deep-learning applications to cross-examine the clinical diagnosis of COVID-19. The applications use some classification models such as multilayer perceptrons, recurrent neural networks, convolution neural networks, etc. to find a prediction accuracy and an area under the ROC curve (AUC). Compared to other classification models, the hyper model so-called long short-term memory with convolution neural network attains an optimized accuracy of ~91% and AUC of ~90%. Khan et al. [27] constructed the expert model based on a deep extreme learning module and artificial neural network to correlate the parameters of coronavirus. The model uses a different number of neurons to describe the nature of activation functions which categorize the medical dataset into several sampling groups to achieve better prediction accuracy. The recent studies relate the research work with the diagnosis of COVID-19 to resolve the issue of a weighted average ensemble (WSM). Wang et al. [27] commutated the electrocardiographic (ECG) pattern of the COVID-19 patient to test the diagnosis of myocardial injury. Li et al. [28] interrogated the death rate of the COVID-19 dataset ~44.24% to materialize the clinical facts of ventricular arrhythmia via a pattern of ECG. Santoro et al. [29] explored the basic electrocardiographic techniques such as rate, rhythm, axis, etc. using QT-prolongation to measure the heart rate activity depicting the risk factor (i.e., ~14% of the COVID-19 patients out of 110).

Sobahi et al. [30] designed a three-dimension convolution neural network (3DCNN) model to classify the image features of COVID-19 using attention mechanism-based deep residual neural network. Rahman et al. [31] explored various deep-learning models to diagnose COVID-19 patients using the relevant ECG images and exploited the behavior of the DenseNet-201 model to inspect the optimistic features of two-class/three-class classification. Therefore, this paper employs a pre-trained deep-learning model which enables self-supervised transfer learning to build an ensemble-based hybrid learning model to offer an effective analysis in the early detection of coronavirus. Table 1 summarizes the significant attributes of the existing frameworks for the early detection of COVID-19.

## 3. Edge-Supported Mobile Computing Systems

The rapid development of mobile communication includes edge computing theory and techniques to bridge the technological gap between the network edges and the cloud. It can speed up the process of content deliveries to improve the quality of networking services (QoNS). It uses system intelligence to drive multimedia services over mobile networks. However, it has experienced more network traffic and device computation to manage the network workloads. The emerging idea evolves the paradigm of mobile edge computing to ease the complexity of a backbone network that has possible storage/computation resources to circumvent the propagation delay [32]. It has a cloud computing infrastructure to support various IoT applications that deal with the proper computation request to mitigate the load factors of the backbone networks. To fulfill the device requirements, the wireless data transmission offloads the computation that might experience the channel congestion of the wireless channels [33]. As a result, the proper decision-making and optimization problems are still addressed to resolve the issues of computation and communication resources of the edge nodes.

Technical strategies such as game theory and convex optimization have been derived for various test cases that include uncertain input, dynamic conditions, and temporal isolation. The uncertain input derives the information factors to reveal the challenges of the wireless channel and security policies. The dynamic condition integrates the computing and communication systems to deal with the resources of the edge nodes. Lastly, temporal isolation considers the Lyapunov optimization to achieve better resource optimization to meet the objective of edge computing systems [34]. However, the complexity of the networks is expected to grow by more than 2000 parameters. To achieve better optimization, it uses the techniques of edge computing that include physical, data-link, and traffic control. To realize the difficulties of edge computing and caching, the communication technologies of IoT such as BLE, Zigbee, and 6LoWPAN are preferred [35]. These technologies develop effective resource management to gather real-time data that may be either distributed or centralized through the knowledge of the learning agents.

### An Intelligent Edge-IoT Model—A Location-Aware System

Cognitive computing builds an intelligent Edge-IoT model to cooperate with computing, communicating, and caching [9]. It has a protocol stack to partition three major parts:

Collecting the information: The edge computing systems sense and collect the observed data through the process of cognitive computing.

Observing the systems: It has a dedicated observation system to perform massive operations and decisional scheduling.

Handling the request: It deals with the requests of the mobility devices to provide a proper decision and scheduling.

The international telecommunication union (ITU) streamlines three scenarios of 5G networks such as ultra-reliable and low-latency communication, enhanced mobile broadband, and massive machine-type communication. These scenarios support diverse services to provide a better network performance in terms of quality of services and quality of experiences. The 5G network emerges various applications namely remote surgery, virtual reality, augmented reality, and holographic projection to connect massive smart devices [36]. To meet the stringent requirements, the enabling architectures revolutionize network intelligence, which includes data rate, latency, reliability, energy efficiency, mobility, and massive connection. It has AI capability to support near-instant and seamless connectivity that includes potential architecture to inspect the space-air-ground-underwater network. The space network co-locates the orbiting satellite to examine the spatial factors, whereas the air network utilizes the aerial platform to provide reliable connectivity; the ground network provides diversified services to exploit the frequency brands, and the underwater network provides surface connectivity to observe the activities of the broad-sea.

The evolution of the 5G network derives the protocols and rules to support the existing architectures such as software-defined networking, network function virtualization, and network slicing. The extended network namely 6G supports high-data rate liability, latency, and connectivity to provide diversified services such as intelligent, adaptive, agile, and flexible. Kato et al. [37] utilized a deep learning technique to optimize the performance of the integrated networks. It has a suitable path to preserve the connectivity of wireless devices. Hence, the AI-enabled intelligent network can be adapted to optimize the network architecture and achieve better network performance. It applies intelligent networking to analyze the sensing data that uses knowledge discovery to provide provisional services such as resource management, network adjustment, and intelligent services. Network automation and high-level intelligence have sensing, data mining, analytics, control, and application to extract significant features from the massive of real-time data. Moreover, the different functionalities include self-configuration, self-healing, and self-optimization to optimize layered design, decision-making, network management, and resource optimization.

Smart Application—It uses specific applications to deliver provisional services that include a fair evaluation to handle intelligent processing. Most of the high-level applications support intelligent programming and management to realize the network features. The main objective is to explore the key factors to examine the quality of services (QoS) that consider the cost dimension to learning the features of the resource efficiencies such as energy efficiency, storage, spectrum utilization, and computation.

Intelligent control—This layer comprises learning, optimization, and decision-making to develop an intelligent agent. It can optimize the computing resources such as routing, spectrum access, power control, and network association to offer diverse services. Most of the resources integrate AI techniques to equip an intelligent system that automates the decision-making process. It has a knowledge-based system that incorporates the learning process to enhance the behavior analysis of any computing device. The evolving network optimizes the network resources in terms of heterogeneity, network slicing, resource management, and PHY layer design to signify the characteristics of the 5G/6G network [38]. AI and 5G/6G networks are integrated to achieve self-organization and self-optimization that employs post-massive multiple input and output to offer thousands of THz transmissions. Moreover, the task optimization applies a rule-based network to discover an auto-learning model that operates the network parameters, resources, and architecture to offer visualization, softwarization, and cloudization. The intelligent agents are cognitive-based to comply with high-quality services that select a suitable spectrum management framework to provide multi-access scenarios.

Mining and Analytics—This layer deals with a massive number of IoT devices to process and analyze the discovery of knowledge data. The massive data collection is typically based on high-dimension, non-linear, and, heterogeneous that applies data mining and analytics to address the challenging issues of any real-time system. Since the raw data is massively collected in a dense network, it highly demands AI-based mining to transform high-dimension into low-dimension space. On the other hand, it may apply data analytics to understand the essential characteristics of wireless networks. To improve the systematic behavior of the 5G/6G network, a valuable pattern can be formulated. It can provide promising solutions such as cloud computing, protocol adaptation, and resource management.

Intelligent Sensing—This layer includes two primary tasks such as sensing and detection to provide intelligent communication via smart devices such as smartphones, drones, and cameras. It can enable massive sensing and detection of environmental data to interface with physical environments such as intrusion detection, frequency utilization, and spectrum sensing. The latest technologies demand real-time and robust sensing to support ultra-low latency and ultra-high reliability. Moreover, it can apply AI techniques to examine the spectrum characteristics to improve sensing accuracy. In order to tackle low-scale and large-scale dimensions, the learning techniques such as ANN, CNN, and RNN are preferred.

Figure 1 illustrates Edge-enabled intelligent eHealth monitoring systems that refer to pervasive healthcare infrastructure [39]. It demonstrates a real-time case study of healthcare services and intensive computation including detection, prevention, response, and recovery. The proposed framework has three-tier components such as data collection, data analytics, data center, and medical diagnosis. The data collection has a built-in structure of IoT subsystems to perform healthcare tasks remotely namely patient monitoring and physiological treatment. The core monitoring system has wireless body area networks to infer the clinical facets namely cough with phlegm, muscle pain, fever, and breathing difficulties. Each system has a dedicated channel between the IoT subsystems and the edge networks. The edge computing networks configure with smart devices such as a tablet, smartphones, smart thermostats, phablets, and smartwatches that have several built-in modules such as medical sensors, application software, and gateway. In addition, the dedicated server connects the IoT devices over the local networks such as WiFi, Zigbee, Bluetooth, and LTE-A to provide interactive communication [40]. Data analytics includes a powerful analysis with AI-enabled functions to carry out intelligent processes such as learning, optimizing, structure organization, and decision-making. The data center is associated with data storage, processing, and analysis to offer on-demand application services. It has potential features including scalability and accessibility to host the virtual sensors, middleware systems, and application services. Importantly, it has three major events (observe, identify, and alert) to analyze and visualize the medical condition of the patient. Lastly, the medical diagnosis offers a data-driven clinical observation that manages the access requests of the doctors and the patients through a monitoring center.

## 4. Proposed Intelligent Edge-IoT Framework

This section discusses the envision of the proposed intelligent IoT framework that monitors, identifies, and analyses the potential threats of coronaviruses in real time. Importantly, it has a prediction model to observe the treatment cases such as supportive care, isolation, and experimental measures. Figure 2 shows the systematic workflow of the intelligent Edge-IoT model that interconnects data collection and uploading, isolation/quarantine center, data observation, and medical experts to provide an active interface through the cloud network [41].

Data Collection and Uploading—The integrative components collect the real-time symptoms using the wearable sensors, implanted in the patient’s body. Most of the COVID-19 symptoms are fever, cough, breathing difficulties, and sore throat, which demands a reliable biosensor to infer the symptoms. For instance, a thermal scanner infers the body temperature; a cough detector integrates acoustic and aerodynamic to measure the forced expiration and throat clearing; the pulse motion interface detects the nature of a weakened state caused by long stress; a novel image processing is utilized to infer the natures of pipelining such as bitter taste, dry cough, and heartburn; and pulse oximetry is a lightweight device to monitor the oxygen level of the blood. It is worth collecting other relevant data such as the traveling and contact history of the patient through a dedicated application device.

Isolation/Quarantine Center—This component records the details of the isolated or quarantined patients at the health center. Each record holds both technical and non-technical data to inspect the symptoms of the patients over time. It may include travel history, contact history, hypertension, diabetes, cardiovascular, and chronic respiratory disease to measure the conditional state of the patients.

Data Observation—The observation center includes proper data analysis and machine learning to build a suitable model that provides a better real-time dashboard for the COVID-19 scenario. The prediction model observes the potential symptoms of COVID-19 to collect and upload in a real-time database. In addition, it may discover some treatment procedure that provides some useful information to address the nature of the diseases.

Medical Experts—The experts monitor the activities of the suspected cases to infer the symptoms indicated. Accordingly, the proposed prediction model is designed to investigate the clinical procedures of the suspected cases. On the other hand, the prediction model may record the possible symptoms of the patients to begin the treatment procedure such as quarantine or hospitalization.

Cloud Networking—This networking infrastructure interconnects real-time entities over the Internet. It allows the patients to: (1) observe and upload the symptom; (2) maintain the medical data over time to derive a specific analysis; (3) communicate the predicted results to the medical experts; (4) store the analyzed information to determine a suitable clinical procedure.

Learning Phase—This phase considers a dataset from Israel Albert Einstein Hospital, Sao Paulo, Brazil which has typical diagnosis information for COVID-19. This dataset is already assessed in [41] to analyze the prediction rate of virus infection using a deep learning approach. To record the medical data systematically, this work considered the workflow of the imaging center. This center was associated with a learning phase to create an efficient learning system that handles imbalanced and other datasets to examine its improved prediction ratio using the classifier systems such as ANN, CNN, and RNN. The systematic flow involves the following steps: (1) collect the original signal from the imaging center; (2) perform the data preprocessing including missing data values, create synthetic data using smote, attribute selection, data scaling, and labeling; (3) apply normalization and segmentation techniques to examine the heartbeat; (4) splitting the data into two sets such as testing and training; (5) establish the appropriate learning methods to examine the workflow; (6) find the optimal parameters to learn the intricacies of the learning models; and (7) evaluate the learning models to observe the experimental results of the matrices as outlined in Figure 2.

Data Description—The results obtained from Albert Einstein Hospital, Sao Paulo, Brazil contains the COVID-19 samples examined during the recurring period of 2020. The samples contain the laboratory results of 111 Nos which hold the information of 5644 Nos patients without gender information.

Data Preprocessing—The collected report has numerous missing values which often occur an error while passing the values as the data input. To address this issue effectively, the features with ~90% missing values were eliminated As a result, the genuine patient records were 600 Nos deriving out of 5644 Nos similar to [23]. To deal with categorical forms and to satisfy numeric inputs, the learning systems used the mathematical equation. Subsequently, the data labeling was used along with the unique number to represent the records in a categorical form defined in the column.

Synthetic Minority Oversampling Technique (SMOTE)—Using this technique, the data imbalance is well addressed while any dataset tries to obtain the desired ratio from the number of data records available in the minority class over the number of data records available in the majority class. On the other hand, compared to undersampling techniques, the oversampling technique always plays a contra-role to improve the category efficiency over the minority samples. Therefore, this technique applies synthetic minority to interpolate the samples using randomly chosen and homogeneous neighboring, which can enhance the rate of detection for the minority samples. However, it has numerous issues to generate a new sample while encountering overlapping, boundary, and noise samples. To address this problem, data cleaning and retrieval are performed in parallel, which eliminates the boundary samples in the minority class to solve the issue of marginal sampling [42].

Recursive Feature Selection (RFS)—Finding the appropriate attributes is so essential to classify the significance of the class. Each class generates its subset of candidates to search the features and subsequently uses them as the input process to evaluate three stages of searching algorithms such as forward, reverse, and insert/remover as the search progress. The forward search includes the individual features one-by-one whereas the reverse identifies the inappropriate or redundant features from the original one. Finally, the insert/remove is performed using randomly generated functions to remove the weakest one [43]. Hence, this paper uses the RFS technique to choose inappropriate features or prune the duplication features which makes the features more predictable to improve generalization and interpretable outcomes.

Self-supervised Transfer Learning (STL) Technique—To perform rigorous testing, this paper conducts its pre-trained practice with the chosen features. In practice, the computing network tries to find the data pattern to perform reliable testing and training through the distribution of the same feature space [44]. The classified dataset carries out a few tasks of classification to reconstruct the network with an appropriate data pattern. This data pattern uses the learning models such as ANN, CNN, and RNN to limit the processing time and improve the learning time with less complexity. Moreover, this learning technique subdues the complex parameters to work with the target domain $TD_{S}$ which discovers a few more parameters such as Learning task $LT_{S}$, subjective domain $SD_{S}$, and resultant learning task $RT_{S}$. To improve the conditional probability of the learning technique, the conditional probability distribution $CP_{D}$ can be expressed as:(1)$$CP_{D}=P_{r}\left(\frac{Y_{T}}{X_{T}}\right)$$

The target domain defines its representation i.e., $TD_{S}=\left\{fs,P_{r}\left(fs\right)\right\}$ into two basic forms:

fs is the designed feature space of the learning technique$P_{r}\left(fs\right)$ is the marginal probability distributed over $f_{s}$ to define the equation as follows:
(2)$$fs=\left\{f_{1},f_{2},f_{3},\ldots ,f_{n}\right\}\;\in \;f_{s}\;$$

In case of any conflicts with $TD_{S}$, the situation cases are as follows:

$f_{s}$ with distinct space recorded as
(3)$$fs_{t}\neq fs_{s}$$$f_{s}$ with distinct marginal distribution as
(4)$$P_{r}\left(fs_{t}\right)\neq P_{r\left(fs_{s}\right)}$$

In addition, $TD_{S}$ represents a specific formula with a task $T=\left\{Y_{L},\left(\;.\right)\right\}$ which has two basic parts:

$Y_{L}$ is a defined labeled space to discern the behavior of the target features$\left(xl_{i},yl_{i}\right)$ is a predictive function to learn the behavior of the training data to illustrate the training set with adequate training samples:(5)$$\left(xl_{i},yl_{i}\right)\;\mathrm{where}\;i\in \left\{1,2,3,\ldots ,N\right\},\;xl_{i}\in fs,\;\mathrm{and}\;yl_{i}\in Y_{L}$$

The probability viewpoint shows the learning feature of $xl_{i}$ as L to observe the equation as:(6)$$L=\left\{Y_{L},P_{r}\left(\frac{Y_{L}}{fs}\right)\right\}$$

In general, if any two distinctive tasks are similar, then the variable space may hold the condition as $Y_{L_{t}}\neq Y_{L_{s}}$. Further, the conditional probability is defined as:(7)$$\left[P_{r}\left(\frac{Y_{L_{t}}}{fs_{t}}\right)\neq P_{r}\left(\frac{Y_{L_{s}}}{fs_{s}}\right)\right]$$

An Optimized Weighted Average Ensemble (OWAE)—Most researchers use a single class model to categorize the classification results. It is worth noting that the ensemble model shows better efficiency than the single-class model to improve the predictive analysis of mining applications [45]. As a result, this paper re-formulates the ensemble model using different modeling algorithms to construct an optimized weighted average ensemble that uses multi-class classification to generate the predictions and to optimize the assigned weighted values using grid search.

Figure 2 shows the proposed intelligent IoT framework that includes three classical observations. They are as follows:

Step 1: The intelligent system noninvasively collects and uploads the medical data including pre-symptomatic and asymptomatic to the diagnostic center.

Step 2: The user may subsequently submit the medical information through the smart application that has the contact information of the infected patients.

Step 3: In addition, the quarantine or isolation center submits the patient’s data periodically to render a proper medical diagnosis and treatment.

Step 4: The data analytics center consolidates the sensed data through the knowledge of smart applications. It can regularize the storage of digital records to infer the intrinsic features of the data using learning techniques. 

Step 5: Upon the inference of data, the medical center presents the analytical information on the real-time dashboard. It may be more informative to the medical experts to construe the clinical procedure for the suspected victims. 

Step 6: If any victim is suspected, the medical expert will contact the patient to undergo clinical trials such as a rapid test kit and polymerase chain reaction. Upon observation, the suspected victims can be identified to track their contact histories.

Step 7: After receiving the laboratory database of COVID-19, the different deep learning models such as ANN, CNN, and RNN evaluate the data to learn, analyze, and interact with complex conditions defined in the STL technique. In ANN, the adaptive neural nodes are connected via a series of sampled data to engineer the basic design of next-generation neurons or to group multiple perceptions in order to process the input in the forward direction. On the other hand, CNN tries to evolve a new detection pattern that applies the sequence of data in a one-dimension form to observe the behavior of the sampling data typically in contrast to machine learning. Finally, RNN generalizes a neural network to examine the data flow via a hidden layer. In order to increase accuracy, healthcare applications rely on DNA sequencing using an iterative ensemble method.

To handle the issue of overfitting, the approaches so-called batch normalization and dropout are employed [46]. Importantly, this paper considers the learning networks such as ANN and CNN to shorten the prediction time and error rate which extend its possible outcome in finding the best computing parameters.

## 5. Results and Discussion

This section discusses the performance metrics of new hyper learning models such as ANN, CNN, and RNN to signify their purpose such as increasing the statistical power eliminating less significant data, and peak-value missing data in resolving the issue of imbalance data. Also, this section explains the result observed using the systematic workflow of the intelligent Edge-IoT model [47].

### 5.1. Learning Analysis

The findings of the laboratory data often range the features with minimum and maximum value (i.e., between 0 and 1). In this study, the models such as ANN, CNN, and RNN were included to examine whether the person had a covid infection or not. Also, the models use the grid search technique to optimize their parameters which is so necessary to increase the statistical power of the computing data. Table 2 defines the significant parameters of the classifier systems. The performance of the learning model applies data splitting to examine the training and testing datasets in the proportion of 80:20. To analyze the learning metrics using ANN, CNN, and RNN, this paper constructed a robust framework with a new pre-trained deep learning model enabled by self-supervised transfer learning [48]. Moreover, this framework utilized the optimistic approach using grid searching to exploit the significant parameters of the classifier systems. These parameters made the system to initialize the core network with the defined neurons. The core network processes one or more inputs to compute the desirable output which is associated with the weights and the feature product to determine its constant (i.e., bias).

Moreover, in the classifier system, a parameter so-called Epoch controls the execution time of the deep learning models over the training dataset. Each model splits the dataset into a small group of batches to eliminate a high data loss as one Epoch has a large number of data feeds to the core network. To minimize the issue of overfitting, the models such as ANN, CNN, and RNN used the dropout rate as an another learning model. This model randomly plunges a few neurons from its hidden layer.

Figure 3 shows the confusion matrices for various learning models including ANN, CNN, and RNN representing the actual and predicted values. The obtained matrices were examined over 10-fold cross-validation which utilizes the learning models with STL to iterate the labels of the samples in order to obtain the optimistic score performing on a new computing data sample [49]. In the cross-validation, the learning models with STL optimized the parameters of the classifier systems not only to increase the overall efficiency but also to provide a better observation on the training data. From the confusion matrices, the true and predicted values were co-examined with the learning models to obtain the computing metrics namely recall or sensitivity, precision, F1-Score, and accuracy. The derivative functions are as follows:

Accuracy—This derivative relates the sufficient information to access the relevant data which can thus classify the instances of unseen data precisely.
(8)$$Accuracy=\frac{T_{P}+T_{N}}{T_{P}+T_{N}+F_{P}+F_{N}}$$Precision—The learning models find the proportionate values of the positive predicted values which are actually held by the positive class.
(9)$$Precision=\frac{T_{P}}{T_{P}+F_{P}}$$Sensitivity or Recall—The learning models determine the proportionate values of the positive prediction which are likely kept out of the predicted records to recall later.
(10)$$Sensitivity\;or\;Recall=\frac{T_{P}}{T_{P}+F_{N}}$$F1-Score—This evaluation metric expresses the performance of the learning models which realize the importance of precision and recall to indicate its high correlation.
(11)$$F1-Score=\frac{\left[2\times recision\times Recall\right]}{\left[Precision+Recall\right]}$$

To evaluate the performance metrics of the learning models including training and testing, a platform so-called Google Colab was chosen. In the course of examination, ADAM as a learning optimizer was employed with a rate of ~0.001. In order to minimize the error rate, the categorical cross-entropy as a loss function was preferred with multi-class classification. Table 3 depicts the performance efficiencies of the learning models with STL. In this observation, the obtained results revealed the significance of the model after preprocessing to analyze the metrics such as accuracy, precision, recall, and F1-Score. The minimum accuracy of the learning models reaches ~94.3% whereas the maximum draws up ~98.3% to show off the effectiveness of the acceptable rating over the number of instances calculated. It is worth noting that the ANN model achieves the optimistic results than other two learning models including CNN and RNN. To balance the computing data highly and to obtain the training and testing accuracy without any loss of data.

Figure 4 shows model loss graph of deep learning models. In this graph, the learning models namely ANN, CNN, and RNN encountered the issue of overfitting via grid searching technique to estimate the number of Epochs. The practical analysis proves that the ANN has a better robustness against the noisy data to assess the complex relationship implicitly among the dependent and independent data varaiables [50]. Also, the ANN effectively trains the dataset to visualize minimum data loss than the other two models such as CNN and RNN. Figure 5 illustrates model accuracy of deep learning model. In representing the accuracy of testing and training data, the deep learning models fit better to analyze the dataset relatively close. Moreover, an efficient trade-off was plotted against accuracy and efficiency to map the applications to process the large dataset.

### 5.2. Network Analysis

A three-tier protocol stack is proposed where the Edge-enabled IoT devices are located at Tier-1 to sense and collect the sensing data; Tier-2 comprises edge computing systems, and Tier-3 deals with data centres to offer storage space and to enable the delivery of analytical results [49,50]. The three-tier protocol stack classifies the selection criteria of the intelligent Edge-IoT model to differentiate the role of each tier. In order to analyze the intelligent Edge-IoT model efficiently, a hardware-based context-aware system was designed using a technological aspect of IoT [51] as shown in Figure 6. The real-time environment considered different experimental scenarios to examine the quality metrics of a smart IoT environment such as power consumption and average response time.

Moreover, this testbed included three basic observations such as collecting, maintaining, and analyzing the clinical data using standard communication technologies such as BLE, Zigbee, and 6LoWPAN. The experimental scenario considered different communication parameters such as network topology, traffic type, and hardware configuration to establish mobility setup and dynamic configuration of the proposed and existing models [49,50]. In this study, the mobility scenario assessed predictive routing and sinks mobility to realize the network movements under the connected devices. The real-time analysis utilized an open-source Python class to develop an intelligent device comprising a sensor, battery, and BLE/Zigbee/6LoWPAN. To carry out the experimental setup, the intelligent device was configured with Rasphian and Arduino. The application server includes a web GUI to analyze the execution of the intelligent device. The web interface accessed the hardware-based system to collect the user information. It has a provision to set up a script that can form a scenario including a number of connected devices, communication modules, network topology, sensing interval, execution time, and system configuration.

To explore the core functionalities of the systems, the web server incorporates the feature of Node.js such as speed and scalability. It may rely on a public movement to prevent the transmission rate of the disease which uses a node registry to perform effective monitoring remotely. In order to examine the variants of the virus, public movements are monitored using GPS coordinates. It may set up a script to build two common request methods including GET and POST. The system controller processes the requested data such as target coordinates to handle the device connectivity using REST APIs. In order to assess the metrics such as power consumption and average response time, the real-time scenario is the target coordinates of the application interfaces including Khan et al. [16] and Al-kahtani et al. [17]. It may include the prone areas of the disease such as human activity, quarantine, and containment zone monitoring to control the infection rate of the disease.

Table 4 defines the key attributes of the networking system. To examine the interconnected devices, three networking scenarios as BLE, 6LoWPAN, and Zigbee were preferred. They utilized a star topology to sense the target coordinates every 5 ms over a period of 45 min. Each networking scenario was reiterated 15 times to detect the mobilities of the target [52]. In order to realize the target data precisely, the connected devices were fully charged during each iteration. As a result, each iteration may precisely collect the consumption of energy to obtain the average consumption of each communication scenario including BLE, Zigbee, and 6LoWPAN.

Figure 7 and Figure 8 show the power consumption ratio 〈%〉 and average Response Time ms versus number of IoT nodes. The real-time scenarios were deployed across the open street to analyse the effects of node positioning in the outdoor environment. The IoT modules such as BLE, Zigbee, and 6LoWPAN were integrated with the functional logics of the proposed IE-IoT, Khan et al. [16], and Al-kahtani et al. [17] to analyse the consumption of power over the transmission of data. To analyze and process the efficiency of modern technologies, this paper primarily focused on the defined topological to vary their target coordinates of BLE, Zigbee, and 6LoWPAN. The context-aware technologies interpreted the sensing data via an intelligent-centric system to evaluate the communication metrics such as power consumption ratio and average response time.

Considering the delivery of data packets, 6LoWPAN proactively handles its code size ~48–128 KB which demands a low routing state to analyze the complexities of the computing nodes including power consumption and transmission delay. Moreover, this technology applies local route repair to minimize the packet overhead, computation, and memory requirements. However, the other wireless technologies such as BLE and Zigbee do not handle the properties of the IoT devices such as latency, routing overhead, dynamic loss, etc. precisely to optimize the control message overhead. The systematic analysis reveals that the proposed IE-IoT module relatively with 6LoWPAN consumes less power consumption and response time than the other two modules [16,17] in terms of three short-range communication technologies such as BLE and Zigbee.

## 6. Conclusions

Detecting the COVID-19 infection at an early stage is crucial for treating the patient on time and controlling the transmission rate of the disease. To lift up the protective measurements, the medical laboratories regulate the clinical test relatively cheaper to carry out the diagnosis and treatment via a proper data management. Moreover, the healthcare systems offer a wide range of clinical procedures not only to maintain the patient records but also to obtain the trials of the biological specimens. Thus, this paper chose a few existing deep learning models such as ANN, CNN, and RNN to predict the disease outbreak based on laboratory results. The learning models utilized a few technical aspects including SMOTE, RFS, and STL to build an ensemble-based hybrid learning model which uses a new pre-trained deep learning model enabled by self-supervised transfer learning to eliminate ineffective data features and to improve the learning outcome with less training time. The performance analysis reveals that the ANN model achieves a better prediction accuracy $\left(98.3\%\right)$ than the other learning models such as CNN and RNN.

In addition, this paper constructed a testbed using Raspbian and Arduino to analyze the critical parameters such as consumption and average response time. In order to observe the critical variants of the people, the mobile sink and remote medical centers assessed the sensitive information of the people using WBAN. The proposed IE-IoT and relevant existing frameworks applied the communication modules such as BLE, Zigbee, and 6LoWPAN to examine power consumption and average response time using a dedicated testbed. The experiment result shows that the proposed IE-IoT with 6LoWPAN consumes less consumption of energy and delay than the other existing frameworks. In the future, the proposed IE-IoT will use a common gateway system which may allow different networking platform to interconnect with computing devices and application interface to offer a location-aware services. Also, this promising solution will significantly cut down the impact of the transmissible disease in the highly populated areas through early-stage detection.

## Figures and Tables

**Figure 1 sensors-23-02995-f001:**
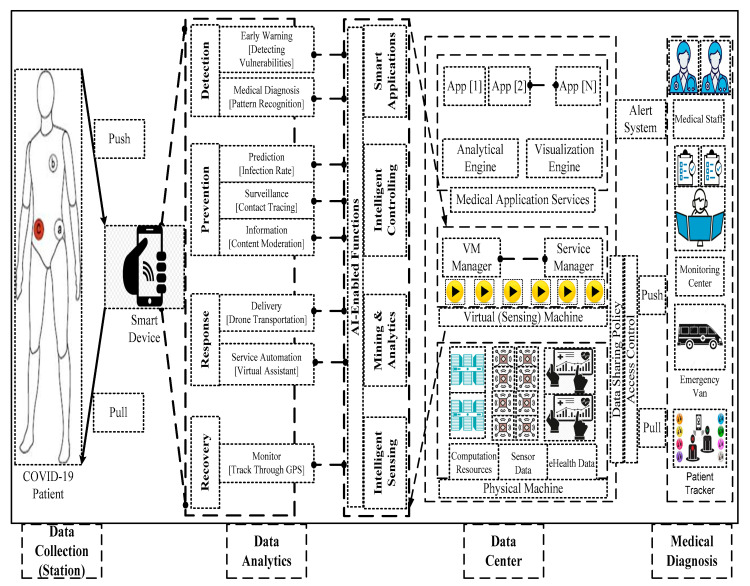
An intelligent Edge-IoT model—a cognitive-aware system.

**Figure 2 sensors-23-02995-f002:**
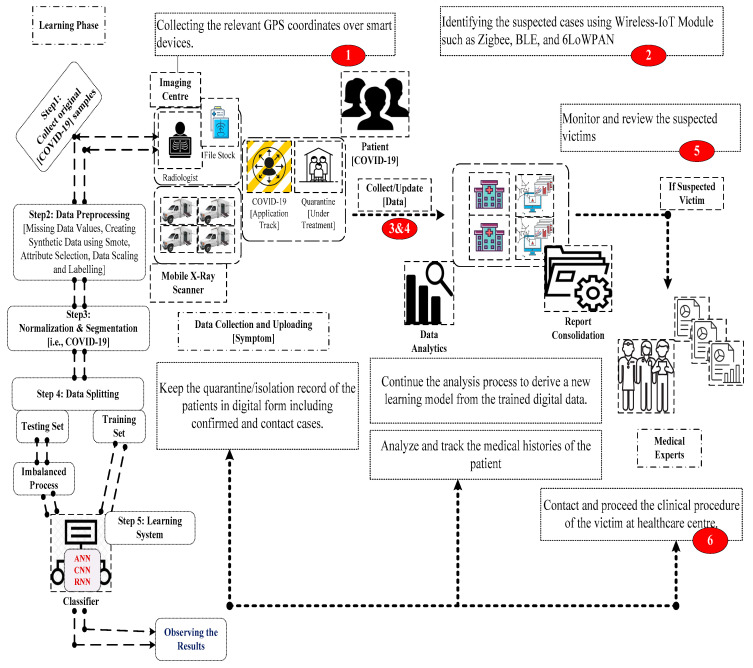
Systematic workflow of intelligent Edge-IoT model.

**Figure 3 sensors-23-02995-f003:**
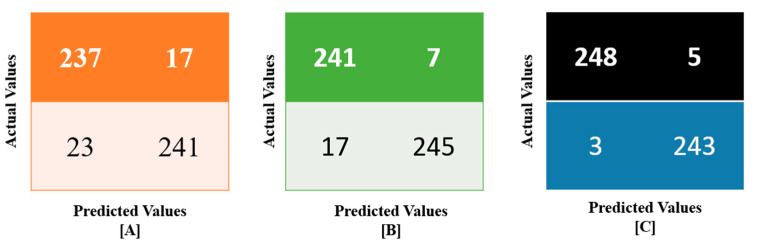
Confusion matrices obtained for various learning models (**A**) ANN (**B**) CNN (**C**) RNN.

**Figure 4 sensors-23-02995-f004:**
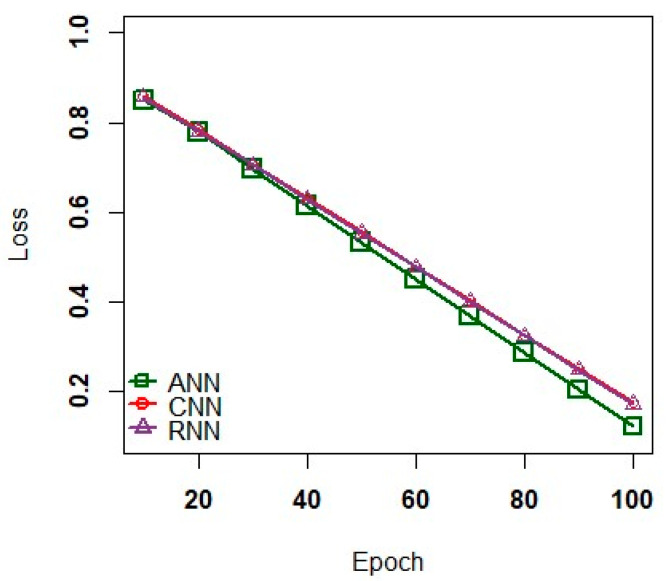
Model Loss versus Epoch.

**Figure 5 sensors-23-02995-f005:**
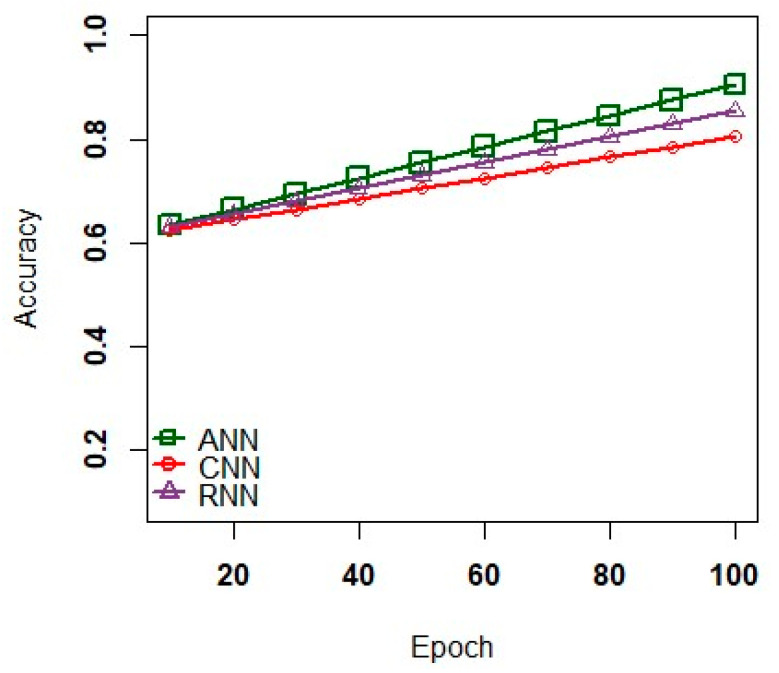
Model Accuracy versus Epoch.

**Figure 6 sensors-23-02995-f006:**
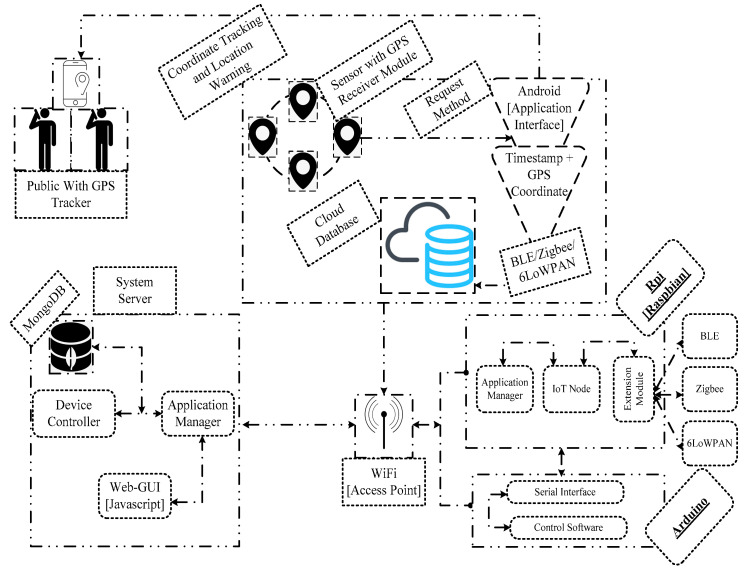
Testbed of intelligent Edge-IoT model.

**Figure 7 sensors-23-02995-f007:**
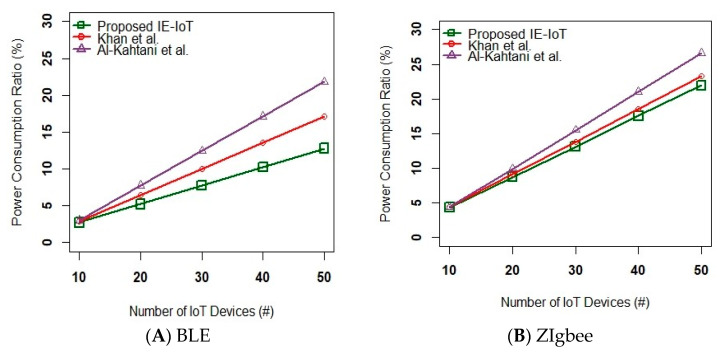
Power Consumption Ratio 〈%〉 versus Number of IoT Nodes [16,17].

**Figure 8 sensors-23-02995-f008:**
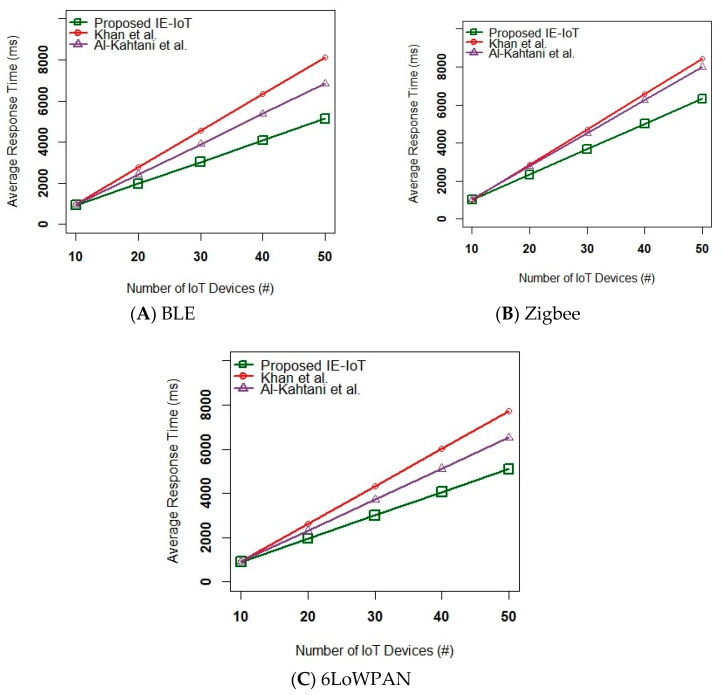
Average Response Time 〈ms〉 versus Number of IoT Nodes [16,17].

**Table 1 sensors-23-02995-t001:** Summarized the significant attributes of the existing frameworks for the early detection of COVID-19.

Frameworks	Applied Dataset	Technique Used	Drawback	Real-Time Analysis
Jiang et al. [25]	RT-PCR Data [Swab]	AI framework with predictive analysis	○ An ensemble-based hybrid learning is not considered. ○ For windows, manual placement was chosen to isolate the interesting region. ○ The obtained dice coefficient is so adverse for the lesion arrangement ○ The multiclass classification technique is not utilized to analyze the tested labeling data	Not Considered
Alakus and Turkoglu [26]	SARS CoV2 Data [Patient Info]	Clinical Predictive Model		Not Considered
Khan et al. [27]	SARS CoV2 Data [Infected Info]	An Optimized Predictive Model		Not Considered
Wang et al. [28]	ECG Images	Retrospective Analysis		Not Considered
Li et al. [29]	ECG Images [Cordio Factor]	General Classification & Analysis		Not Considered
Santoro et al. [30]	ECG Images [Cardio Factor]	General Classification & Analysis		Not Considered
Sobahi et al. [31]	ECG Images [CT Scan & X-ray]	Three Dimension Convolution Neural Network		Not Considered

**Table 2 sensors-23-02995-t002:** Significant parameters of the classifier systems.

Parameters	ANN	CNN	RNN
Number of neurons in the learning networks	64, 32, 16, 8	64, 32, 16, 8	256, 128, 64, 32
Number of computing layers	1, 2, 3, 4	1, 2, 3, 4	1, 2, 3, 4
Activation Function	Exponential Linear Unit (ELU)	Exponential Linear Unit (ELU)	Exponential Linear Unit (ELU)
Applied Loss Function	Categorical Cross Entropy (CCE)	Categorical Cross Entropy (CCE)	Categorical Cross Entropy (CCE)
Number of Epochs	Optimum ~100	Optimum ~100	Optimum ~20
Learning Optimizer	Adam	Adam	Adam
Dropout	0.25	0.15	0.20
Batch Size	15	24	55

The loss and cost functions of CCE are as follows: $Loss=-\overset{K}{\underset{L=1}{\sum }}Y_{L}.log\hat{\left(Y_{L}\right)}$ where K is the number of classes available in the sampling data; $Cost\;\frac{1}{N}\overset{N}{\underset{i=1}{\sum }}\overset{K}{\underset{L=1}{\sum }}\left[Y_{iL}.log\left(\hat{Y_{iL}}\right)\right]$ where K is the sampling class; $Y_{iL}$ is the active value of the distinct task; and $\hat{Y_{iL}}$ is the prediction value of the neural network.

**Table 3 sensors-23-02995-t003:** Performance efficiency of the learning models with STL.

Learning Models	$\mathrm{ANN}\;\left(\%\right)$	$\mathrm{CNN}\;\left(\%\right)$	$\mathrm{RNN}\;\left(\%\right)$
Accuracy	98.3	96.7	94.3
Precision	97.1	95.3	94.1
Sensitivity or Recall	99.1	97.2	98.3
F1-Score	98.1	96.3	93.7

**Table 4 sensors-23-02995-t004:** Key Attributes of the Networking System.

Key Attributes	Values
Number of Interconnected Devices	50 Nos.
IoT Communication Modules	BLE/Zigbee/6LoWPAN
Scenario	GPS Coordinate
Duration	45 mins
Transmission Interval	5 ms
Data Packet	6 Bytes [Latitude and Longitude]
Setup Node	Static
Topology	Star
Number of Iteration	15 times

## Data Availability

Not applicable.

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
