# Peer review of "EEI-IoT: Edge-Enabled Intelligent IoT Framework for Early Detection of COVID-19 Threats"

_sensors, 2023, doi:10.3390/s23062995_

Round 1

Reviewer 1 Report

The authors performed COVID-19 detection based on Edge-Enabled Intelligent IoT Framework. The made some good efforts but there are some serious concerns which needs to be addressed carefully:-

1. The rise of respiratory infections (SARS, SWIN flu, and COVID19) over the last two decades has made it clear that early detection is critical to avoiding a pandemic catastrophe. COVID was detected in 2019, so you can’t write the confusing statement.

2. Use the identical terms throughout the manuscript i.e. covid19, COVID-19. COVID19.

3. There are various acronyms are used without the full form on its first appearances i.e. BLE, 6LoWPAN, SARS, etc. Check and write the full forms used in the whole manuscript.

4. There is recommended to add motivation and contribution statement at the end of introduction section.

5. The Section Related Work is recommend to add with some recent literature citations.

6. After the “Data collection and uploading” subsection, it is recommended to add one more subsection “Preprocessing, Feature Selection, and Normalization” and the relevant details in it.

7. I believe there should be Machine Learning framework for automatic detection of Covid-19 based on collected data, as manual monitoring is not the best solution in this case.

8. Follow the given studies (10.1155/2022/7713939, 10.3390/s22207722)

9. Recommended to add confusion matrix diagram to give better understanding of the predicted results.

10. The comparative analysis of recent methods may be added.

11. There isn’t any latest study of 2022 has been cited, some latest literature of 2022.

Author Response

Comment 1: The rise of respiratory infections (SARS, SWIN flu, and COVID19) over the last two decades has made it clear that early detection is critical to avoiding a pandemic catastrophe. COVID was detected in 2019, so you can’t write the confusing statement.

Response 1: As you advise, we have carefully read and incorporated the reasonable changes in the abstract of the paper to avoid the conflict of the statement.

Coronavirus Disease 2019 (COVID-19) has caused severe acute respiratory syndrome coronavirus 2 (SARS CoV 2) across the globe recommended effective diagnosis and treatment for any chronic illnesses and long-term health implications. In the worldwide crisis, the catastrophic plague of this pandemic shows its daily extension (i.e., active cases) and genome variants (i.e., Alpha) within the virus class to diverse the association with treatment outcomes and drug resistance.

You are in earnest asked to see Page No.1.

Comment 2: Use the identical terms throughout the manuscript i.e. covid19, COVID-19. COVID19.

Response 2: As you advise, we have carefully read and incorporated the reasonable changes in the abbreviated terms including COVID-19 to improve the readability of the paper.

You are in earnest asked to see Page No.1-20.

Comment 3: There are various acronyms are used without the full form on its first appearances i.e. BLE, 6LoWPAN, SARS, etc. Check and write the full forms used in the whole manuscript.

Response 3: As you advise, we have carefully read and incorporated the reasonable changes in the form of tabulation defining the various acronyms used in this paper to improve the readability of the paper.

You are in earnest asked to see Page No.20.

Comment 4: There is recommended to add motivation and contribution statement at the end of introduction section.

Response 4: As you advise, we have carefully read and incorporated the reasonable changes including Motivation and Contribution in Section 1 [Introduction] to the objectives of this paper.

You are in earnest asked to see Page No.2-3

Comment 6: After the “Data collection and uploading” subsection, it is recommended to add one more subsection “Preprocessing, Feature Selection, and Normalization” and the relevant details in it.

Response 6: As you advise, we have carefully read and incorporated the reasonable changes in adding a few more subsections to discuss the core aspects including preprocessing, feature section, etc. Also, we have reconstructed the learning phase to anlayze the deep learning such as ANN, CNN, and RNN to estimate the prediction accuracy of the covid dataset.

You are in earnest asked to see Page No.10-12

Comment 7: I believe there should be Machine Learning framework for automatic detection of Covid-19 based on collected data, as manual monitoring is not the best solution in this case.

Response 7: As you advise, we have carefully read and incorporated the reasonable changes in analyzing the sampled data using deep learning as well as inspecting the network in real-time using a dedicated testbed. Also, we re-developed the recent state-of-the-art frameworks along with the proposed one to evaluate the system parameter such as power consumption and average transmission delay.

You are in earnest asked to see Page No.13-16.

Comment 8: Follow the given studies (10.1155/2022/7713939, 10.3390/s22207722)

Response 8: As you advise, we have carefully read the suggested articles and incorporated the reasonable changes in developing and analyzing the sampled data of COVID-19 using deep learning.

You are in earnest asked to see Page No.1-20

Comment 9: Recommended to add confusion matrix diagram to give better understanding of the predicted results.

Response 9: As you advise, we have carefully read and incorporated the evaluated matrices in Section 5. Also, we re-formulated the structure of the learning models using self-supervised transfer learning to examine their prediction accuracy.

You are in earnest asked to see Page No.14

Comment 10: The comparative analysis of recent methods may be added.

Response   10: As you advise, we have carefully read and incorporated the comparative analysis in Section 5.1 to highlight the significance of the proposed IE-IoT in comparison with other state-of-art approaches.

You are in earnest asked to see Page No.14-16

Comment   11: There isn’t any latest study of 2022 has been cited, some latest literature of 2022.

Response    11: As you advise, we have carefully read and incorporated the recent articles in Section 1 and Section 2 to strengthen the scope this progress.

You are in earnest asked to see Page No.1-20

Reviewer 2 Report

The article lacks novelty, and the author's contributions are not highlighted in the manuscript.

Simulation results are insufficient to validate the proposed model.

Why does the BLE algorithm perform better than the others? Provide the reasons.

Please provide detailed comparisons to existing state-of-the-art techniques.

Your contributions regarding the early detection of COVID-19 are missing from the manuscript. Please elaborate on your contributions using the results obtained.

In addition to the star topology, other topologies may also be used in simulations.

Author Response

We would like to thank the reviewers and the editor for their valuable suggestions and possible recommendation towards the publication. To facilitate the review process further, we have carefully read and addressed the reviewer’s comments, typos, punctuation, and grammatical errors.

Below is a one-to-one response to the reviewers’ comments on the re-submitted revised article [sensors-2053803].

Comment 1: The article lacks novelty, and the author's contributions are not highlighted in the manuscript.

Response   1: As you advise, we have carefully read and incorporated the reasonable changes in Section 1 to highlight the major contribution of this paper.

You are in earnest asked to see Page No.3.

Comment 2: . Simulation results are insufficient to validate the proposed model.

Response   2: As you advise, we have carefully read and incorporated the reasonable changes in Section 5 to demonstrate the significance of the proposed IE-IoT with other state-of-the-art approaches.

You are in earnest asked to see Page No.13-19.

Comment  3: . Why does the BLE algorithm perform better than the others? Provide the reasons

Response   3: As you advise, we have carefully read and incorporated the reasonable changes in Section 5.2 to demonstrate the significance of the proposed IE-IoT with other state-of-the-art approaches.

You are in earnest asked to see Page No. 19.

Comment  4: . Please provide detailed comparisons to existing state-of-the-art techniques.

Response   4: As you advise, we have carefully read and incorporated the reasonable changes in Section 2 to show a comparative analysis with the recent state-of-the-art approaches.

You are in earnest asked to see Page No. 5.

Comment  5: . Your contributions regarding the early detection of COVID-19 are missing from the manuscript. Please elaborate on your contributions using the results obtained.

Response   5: As you advise, we have carefully read and incorporated the reasonable changes in Section 5.2 to demonstrate the significance of the proposed IE-IoT with other state-of-the-art approaches.

You are in earnest asked to see Page No. 13-16.

Reviewer 3 Report

PFA

Author Response

We would like to thank the reviewers and the editor for their valuable suggestions and possible recommendation towards the publication. To facilitate the review process further, we have carefully read and addressed the reviewer’s comments, typos, punctuation, and grammatical errors.

Below is a one-to-one response to the reviewers’ comments on the re-submitted revised article [sensors-2053803].

Comment 1: What is the novelty of the proposed work? Describe the details in the abstract.

Response   1: As you advise, we have carefully read and incorporated the reasonable changes in Section 1 to highlight the major contribution of this paper.

You are in earnest asked to see Page No. 3, 13

Comment 2: Introduction should be updated with the specific contributions and objectives focused on the paper. A single reference is not much enough for the introduction, it is very important to cite numerous recent works published in 2020, 2021, and 2022 for the problem identification and

motivation.

Response   2: As you advise, we have carefully read and incorporated the reasonable changes in Section 1 to address the comments such as contributions and objectives.

You are in earnest asked to see Page No. 2-3.

Comment 3: Also, the authors should review some of the existing works associated to this study with the challenges, issues, and suggestions.

Response   3: As you advise, we have carefully read and incorporated the reasonable changes in Section 2 to address the issues of the existing works.

You are in earnest asked to see Page No. 4-5.

Comment  4: The descriptions for the proposed work is very less, it could be highly difficult to understand the workflow and model. Provide a complete and detailed explanation for Figure 1 and Figure 2.

Response  4: As you advise, we have carefully read and incorporated the reasonable changes in Section 3 and 4 to discuss their working principles.

You are in earnest asked to see Page No.8-10.

Comment 5: . In Section 4, the author mentioned that novel image processing is utilized to 216 infer the natures of pipelining such as bitter taste, dry cough, and heartburn; and pulse 217 oximetry is a lightweight device to monitor the oxygen level of the blood”. What is the methodology? Explain in etail.

Response  5: As you advise, we have carefully read and incorporated the reasonable changes in Section 4 to discuss their learning procedures with metric evaluation in detail.

You are in earnest asked to see Page No.10-12

Comment 6: In results and discussion, the authors have used only one graph to validate the proposed framework. By using only a single parameter, how the authors state that the EEI-IoT framework results better?

Response 6: As you advise, we have carefully read and incorporated the reasonable changes in Section 6 to validate the proposed model with other state-of-the-art appraoches. Also, we reconstructed the learning models to analyze their key features in terms of accuracy, recall, precision, etc.

You are in earnest asked to see Page No.12-16

Comment  7: The results section should be more elaborated with different measures, and an extensive comparative must be presented to demonstrate that the suggested model outcomes better.

Response  7: As you advise, we have carefully read and incorporated the reasonable changes in Section 5 to demonstrate the significance of the proposed model with other state-of-the-art approacges as discussed in Comment 4.

You are in earnest asked to see Page No.12-16.

Comment 8: Conclusion is very short and not good. It must be rewritten with proper findings, contributions, attainments, limitations, and future scope..

Response 8: As you advise, we have carefully read and rewritten the conclusion section i.e,. Section 6 to address the findings, attaintment, limitation, and future scope.

You are in earnest asked to see Page No.22-23

Comment  9: The number of references 15 are not enough, should add more recent references in 2021 and 2022. Consider the recent reference: Predicting Epidemic Outbreaks usi The Fusion of Internet of Things, Artificial Intelligence, and Cloud Computing in Health Care, Internet of Things, Springer, vol. 1, Issue 1, 2021, pp. 197-222.

Response  9: As you advise, we have carefully read and incorporated the reasonable changes in this revised draft.

You are in earnest asked to see Page No.1-23.

Round 2

Reviewer 1 Report

The authors have made significant efforts to incorporate all the suggested changes. Now I would recommend this publication.

Reviewer 2 Report

The author's comments are in the form of paragraphs, and it is difficult to find the author's response. 

The rest of the authors responded to all the comments, and the revised version is much improved. 

Reviewer 3 Report

The authors have incorporated the suggestions, and the paper can be accepted in its current form.